# Digital Health Profile of South Korea: A Cross Sectional Study

**DOI:** 10.3390/ijerph19106329

**Published:** 2022-05-23

**Authors:** Kyehwa Lee, Libga Seo, Dukyong Yoon, Kwangmo Yang, Jae-Eun Yi, Yoomi Kim, Jae-Ho Lee

**Affiliations:** 1Department of Information Medicine, Asan Medical Center, Seoul 05505, Korea; geffa@amc.seoul.kr (K.L.); rod3645@naver.com (L.S.); 2Department of Biomedical Systems Informatics, Yonsei University College of Medicine, Yongin-si 16995, Korea; dukyong@yuhs.ac; 3Center for Health Promotion, Samsung Medical Center, Sungkyunkwan University School of Medicine, Seoul 06351, Korea; kwangmo@skku.edu; 4Korea Health Information Service, Seoul 04515, Korea; joy@k-his.or.kr (J.-E.Y.); ymkim@k-his.or.kr (Y.K.); 5Department of Emergency Medicine, Asan Medical Center, University of Ulsan College of Medicine, Seoul 05505, Korea

**Keywords:** health information system, information and communication technology, electronic medical records, personal health records, clinical data warehouse

## Abstract

(1) Backgroud: For future national digital healthcare policy development, it is vital to collect baseline data on the infrastructure and services of medical institutions’ information and communication technology (ICT). To assess the state of medical ICT across the nation, we devised and administered a comprehensive digital healthcare survey to medical institutions across the nation. (2) Methods: From 16 November through 11 December 2020, this study targeted 42 tertiary hospitals, 311 general hospitals, and 1431 hospital locations countrywide. (3) Results: Since 2015, most hospitals have implemented electronic medical record (EMR) systems (90.5 percent of hospitals, which is the smallest unit, and 100 percent of tertiary hospitals). The rate of implementation of personal health records (PHRs) varied significantly between 61.9 percent and 2.4 percent, depending on the size of the hospital. Hospitals have implemented around three to seven government-sponsored information/data transmission and receiving systems for statistical or investigative objectives. For secondary usage of medical data, more than half of tertiary hospitals have implemented a clinical data warehouse or shared data model. However, new service establishments utilizing modern medical technologies such as artificial intelligence or lifelogging were scarce and in the planning stages. (4) Conclusion: This study shows that the level of digitalization in Korean medical institutions is significant, despite the fact that the development and spending in ICT infrastructure and services provided by individual institutions imposes a significant cost. This illustrates that, in the face of a pandemic, strong government backing and policymaking are essential to activate ICT-based medical services and efficiently use medical data.

## 1. Introduction

The worldwide coronavirus disease of 2019 (COVID-19) was a harsh illustration of human survival, with varied morbidity and fatality rates due to each country’s government’s choices and information and communication technology (ICT) state [1,2]. Prevention and quarantine strategies and methods varied per country, based on its culture and quality of medical infrastructure. COVID-19’s high infection rate resulted in a quick increase in patient numbers, placing an undue load on healthcare systems and ultimately resulting in the collapse of the system’s weakest link, demonstrating a pattern of rapidly growing mortality [3,4]. Effective infection control requires clear and accurate information exchange and rapid decision-making between policymakers, medical personnel, and the general public via ICT [5]. As a result, it is now widely recognized that healthcare system informatization has become a crucial determinant of the quality of national medical services. Evaluating a country’s health and medical information technology standing is challenging. Through the Health Metrics Network, the World Health Organization (WHO) has supplied diagnostic tools for health and medical information to impoverished and emerging countries since 2008 [6]. Additionally, the WHO published an atlas summarizing the eHealth status of 125 member countries based on the eight areas covered in the 2015 global survey. The European Union (EU) benchmarked the availability and use of eHealth by general practitioners (GPs) in 27 EU member states in 2018 through the benchmarking of eHealth deployment among GPs. While overall adoption of eHealth among primary care physicians grew, significant disparities in adoption rates across the nations surveyed [7]. GPs frequently used eHealth in countries with the most considerable adoption rates (Denmark, Estonia, Finland, Spain, Sweden, and the United Kingdom), while the eHealth system was not widely adopted in the nations with the lowest adoption rates (Greece, Lithuania, Luxembourg, Malta, Romania, and Slovakia) [8]. To promote the use of high-quality electronic medical record (EMR) systems and assess the state of healthcare information technology (IT) in South Korea, the Ministry of Health and Welfare conducted a countrywide survey in 2015 and 2017. However, the previous two surveys concentrated exclusively on hospital EMR systems. Other medical information systems, such as order communication systems (OCS), personal health records (PHRs), picture archiving and communication systems (PACS), laboratory information management systems (LIMS), and clinical data warehouses (CDWs), were omitted. South Korea has become a global leader in ICT [9], and as part of its preparations for the Fourth Industrial Revolution, the country is promoting national health and medical informatization in order to provide safer and higher-quality medical services anytime and anywhere through the use of big data in the medical field. This study was constructed with a broader and more diverse survey scope in mind, with the goal of determining the explicit level of informatization in South Korean hospitals nationally. Through in-depth research, the data from this survey will aid in the development of a national health and medical informatization strategy plan and the development of informatization indicators for medical institutions.

## 2. Materials and Methods

This research aimed to examine the medical ICT infrastructure and service status of as many hospitals as feasible. To this purpose, we intended to go beyond the limits of the national EMR system building status surveys conducted in 2015 and 2017. IT systems to facilitate secondary use of medical data, additional systems beyond EMR, such as PHR and linkage with government surveillance systems, the status of medical services leveraging emerging technologies such as artificial intelligence, etc., were all newly included items in the 2020 survey. Expert consensus established the scope of the survey so that the results might serve as fundamental data for future health information governance and policy development. The study was conducted in stages, beginning with establishing the scope of the hospital to be surveyed, constructing a survey by field, running a prototype survey and getting comments, confirming the final survey questionnaire, and then completing the actual survey.

### 2.1. Study Design

The Korea Health and Medical Information Service, which is overseen by the Ministry of Health and Welfare, designed this survey in collaboration with related societies such as the Korean Society of Medical Informatics (KOSMI), the Korean Hospital Information Association (KHIA), and the Korean Health and Medical Information Managers Association (KHMIMA), in order to incorporate expert opinions and improve the survey’s quality. Although it was based on previous surveys conducted in 2015 and 2017, it was expanded to encompass a broader range of themes, including standardization and security of medical data, integration of disparate systems and data, secondary data utilization, workforce development, and IT system governance. As a preliminary stage, we categorized the topics into four categories: fundamental eHealth status, hospital eHealth quality, advanced healthcare information technology system, and eHealth system for secondary usage. As a primary questionnaire, a draft survey consisting of 141 items was produced by numerous advisory panels, including informatics professionals. A pilot survey was conducted in six tertiary hospitals to assess the questions’ reliability, their difficulty level, and the survey method’s optimization. The final questionnaire was amended based on a review of individuals surveyed and focused group interviews conducted during the pilot survey. Unnecessary items were eliminated to alleviate respondents’ burdens and boost response rates. The questionnaire design process is illustrated in Figure 1, and the subjects of the final questionnaire, which contains 94 items organized into four categories, are listed in Table 1.

### 2.2. Survey Target Hospitals and Sampling

The 2009 Medical Act classifies medical institutions in South Korea according to their size, facilities, equipment, and human resources (Table 2). We intended to cover all 42 tertiary institutions and 311 general hospitals located around the country. In total, 217 hospitals were chosen as the target from 1431 hospitals using a sampling procedure that classified all hospitals into 17 regions, sorted them by the number of claims they received, and then sampled them. Additionally, by segmenting the questionnaire items by hospital level, questionnaires for high-level issues such as secondary utilization and smart hospitals were administered selectively to hospitals with more than 300 beds.

### 2.3. Conducting a Survey

Due of the questionnaire’s extensive reach and depth, responding to the survey was challenging. To address this obstacle, we used two distinct survey methodologies deliberately. The KHIA and KHMIMA collaborated on the field investigation for the large-scale hospitals (more than 300 beds). Interviewers were trained by medical information professionals from these organizations. In addition, two online briefing sessions were held for respondents participating in the research response at each hospital. With fewer than 300 beds, Gallop Korea used a combination of survey methods, including online surveys, fax/e-mail surveys, and in-person interviews (using tablet computers or paper questionnaires). The study design and pilot survey were carried out from September to October 2020, and the online briefing session was held on 16 November 2020. The final survey was conducted from 16 November 2020 to 11 December 2020, with some supplementary surveys continuing until 31 December 2020.

## 3. Results

After reviewing the total response rate, we subdivided and examined the response findings based on the primary scope of the survey. There were numerous complex inquiries in four areas, and the investigation’s extent varied based on the size of the hospital; therefore, we separated the results into six subjects and discussed each one separately to facilitate comprehension.

### 3.1. Response Rate

Tertiary hospitals, general hospitals with more than 300 beds, and general hospitals with fewer than 300 beds were included in the sample, with 42, 124, 187, and 217 hospitals, respectively. As a result, the poll received responses from 100%, 90%, and 116.1% of respondents from tertiary hospitals, general hospitals, and hospitals, respectively.

### 3.2. Main IT Systems for Clinical Practices

Three key information technology systems were used in clinical practice: an EMR to track and record patient data, an OCS to communicate orders, and a PACS to manage picture data. In our survey, the EMR, OCS, and PACS implementation rates in tertiary hospitals had remained constant since 2015, whereas the EMR adoption rate in general hospitals had increased by 5.4% (90.6% to 96.0%) since 2015. On the other hand, OCS and PACS installations in general hospitals have fallen marginally since 2015, from 100 to 96.4 percent for OCS and 99.0 to 97.4% for PACS. Both EMR and PACS adoption rates grew in hospitals, from 75.9% to 90.5% and 72.2% to 87.3%, respectively. However, there was no substantial difference in OCS deployment (85.2% vs. 85.1% ). Figure 2 depicts the evolution of each system’s construction rate over time. Apart from the systems required for clinical practice, IT subsystems in hospitals were primarily used to support specialized departments or data management functions. The adoption rate of two representative systems was investigated: an in-hospital pharmacy management system (PMS) that manages the prescription and introduction of drugs used in hospitals and clinical pathology, and a laboratory information management system (LIMS) that manages specific laboratory test information and data. Since 2015, the implementation rate of PMS has increased from 78.9 to 90.5%, 66.7 to 76.3%, and 30.0 to 44.5% in tertiary hospitals, general hospitals, and hospitals, respectively. Figure 2 depicts the evolution of the construction rate of major hospital information technology systems over time. The replies indicated that the rate of LIMS adoption had not increased constantly since 2017.

### 3.3. PHR Use and Functionality

In terms of hospital-tethered mobile PHR systems, 61.9% of tertiary hospitals, 22.6% of general hospitals with 300 beds or more, 4.4% of general hospitals with less than 300 beds, and 2.4% have implemented one. We determined the primary functions of hospital-provided PHR. Online appointment and payment were the most often available PHR capabilities, with 69.0% and 59.5% of tertiary hospitals, respectively, providing them. Additionally, 64.3% of tertiary hospitals provided the right to monitor a patient’s medical information and test results via PHR. However, the frequency with which actual clinical information was downloaded or with which the capacity to get legal documents was provided was relatively low (Table 3).

### 3.4. Health Information Exchange

Transfers of medical information can be classified as hospital-patient or hospital-hospital. Hospitals provided medical information to patients in a number of ways, the bulk of which were paper printouts. Additionally, the rate of storage and distribution of removable media such as CDs or USB drives was high, indicating that 65.8% of tertiary hospitals and 45.5% of hospitals provide medical information to patients using removable media. Hospitals that implemented PHR provided patients with test results and some clinical documents via PHR, but these clinical data were not shared between hospitals. In South Korea, there were two distinct routes for patients to be referred from a low-level hospital to a higher-level hospital. One was a referral system developed by the Health Insurance Review and Assessment Service for claim submissions (HIRA). This system can store patient information and transfer it with higher-level institutions based on the severity of the patient’s illness. Additionally, the transfer fee is calculated and paid to the hospital delivering the service based on this information. The other is the Ministry of Health and Welfare’s medical information exchange system. This system is intended to facilitate the sharing of patient data in accordance with industry standards, such as the Health Level 7 clinical document architecture [10]. These two technologies are now in the pilot phase and will simplify the exchange of patient information using national standards. The establishment of the HIRA medical institution referral system occurred at a rate of 85.7%, 56.7%, and 43.3% in tertiary hospitals, general hospitals, and hospitals, respectively, while the establishment of the treatment information exchange system occurred at a rate of 78.6%, 39.7%, and 18.7% in tertiary hospitals, general hospitals, and hospitals, respectively.

### 3.5. Systems to Support Data Transfer to the Central Surveillance System

In South Korea, the Ministry of Health and Welfare and the Korea Centers for Disease Control and Prevention manage medical data use systems for a variety of reasons, including enhancing national medical safety and early disease detection. These systems assist hospitals in making decisions by gathering unique data from each hospital in real time or retrospectively, responding promptly to health risk situations, and conducting long-term analysis to policymaking. This topic was first evaluated in the 2020 survey. Drug utilization review (DUR) services, opioid and narcotics use management systems (ONMS), and the Korea Adverse Event Reporting System are all examples of representative drug use systems (KAERS). Additional surveillance systems exist, including the Infectious Disease Reporting Support Program (IDRS), the vaccination registration management system (VRS), the national emergency medicine information network (NEMIN), and the life-sustaining medical information processing system (SDLST). Some collect clinical data via their data portal, while the remainder are directly connected to hospital EMRs. As illustrated in Figure 3, the larger the hospital, the faster the system establishes with the government investigative system.

### 3.6. Advanced IT Systems for Secondary Use

As more patients flock to South Korea’s larger hospitals, these larger hospitals are acquiring more advanced clinical data utilization systems for secondary use. Strengthening their research capacity not only enhances their knowledge and practice, but is also viewed as a valuable strategy for establishing a dominant position among hospitals in South Korea’s highly competitive environment. To investigate this phenomenon, we evaluated IT systems that support the use of clinical data for secondary purposes. Three well-known research information technology systems exist: CDW, a clinical data warehouse built within hospital units; CDM, a common data model for multicenter clinical research; and CTMS, which assists in the conduct and maintenance of clinical trials. CDW, CDM, and CTMS all had establishment rates greater than 50% in tertiary hospitals, with CDM having the highest rate (64.3%). However, when asked about future plans, it was discovered that the establishment of research information systems in general hospitals was low (approximately 10%), as illustrated in Figure 4.

### 3.7. Future Healthcare IT Infrastructure

We evaluated how and how much each of the five fundamental technologies–internet of things (IoT), big data, mobile, artificial intelligence (AI), and fifth-generation technological standard (5G)–was used in each hospital and their future investment plans. These five technologies were included in the survey because the Korean Ministry of Health and Welfare selected them as new technologies requiring active investment for the future transition to digital medicine. To confirm the usage status of these technologies in-depth, we investigated how these technologies are being used or are planned to be used for eight major services derived from expert consensus. The survey asked respondents about their experiences with these technologies in eight distinct service areas: (1) Communication between medical staff and patients, (2) Medical service design and resource utilization, (3) Medication error detection, (4) AI-based decision support mode, (5) AI-based digital therapeutics, (6) Early confrontation to infectious crisis, (7) At-risk and emergency patients tracking, and (8) Hazard alert and monitoring. The utilization status of each technology for each service was evaluated and categorized as “active use” or “action plan”, and the findings were analyzed by dividing by the hospital’s size. Table 4 summarizes these domains and the extent to which the five key technologies are applied. Four of the eight services confirmed an instance of a differentiating characteristic for five technologies: Communication between medical staff and patients (Communication), AI-based decision support model (Decision support), Medication error detection (Error detection), At-risk and emergency patients tracking (Patient tracking). As indicated in Figure 5, there are significant disparities between the sorts of services and the fundamental technologies believed to be required to supply these services. Specifically, expected benefits such as the active use of mobile technologies for communication and IoT technology for patient tracking were demonstrated. In addition, it was established that big data and artificial intelligence technology displayed a close correlation. According to respondents, mobile technology is the most critical technology for communication services, while AI is the most core technology for decision support services; AI and big data are the most pivotal technologies for error detection, while IoT is the most powerful technology for patient tracking, as illustrated in Figure 5. This attitude, however, varied according to hospital size; opinion about the importance of mobile technology was generally more favorable in general hospitals than in tertiary hospitals.

## 4. Discussion

### 4.1. Summary of the Survey

This study summarized the most comprehensive and diverse survey on the current state of medical information technology conducted to date in a single country. According to this study, more than 90.5% of hospitals in South Korea have implemented EMRs on average, and 42 tertiary hospitals have maintained a 100% EMR implementation rate since 2015. In contrast to a 2016 study that found that EMR adoption rates were higher in primary care (81%) than in hospitals for inpatient care (76%) in OECD countries, higher-ranking hospitals in South Korea demonstrated a higher rate of EMR adoption. South Korea demonstrated the highest level of medical IT infrastructure status when compared to WHO surveys. The purpose of this study was to measure the rate of implementation of various data repository systems, including CDW, CDM, and CTMS, for secondary use of clinical data, and this was the first attempt to examine clinical research system use on a national scale. It was discovered that more than half of tertiary hospitals had implemented at least one of the research support systems that facilitate secondary use of clinical data. Furthermore, this study included a survey on new medical services utilizing advanced IT technologies such as IoT and 5G technology for the first time. While there were differences between institutions, it was established that hospitals were attempting to implement a variety of ICT services.

### 4.2. The Korean Medical System’s Distinctive Features

South Korea has a government-healthcare system that regulates medical personnel qualifications, as well as the establishment, operation, and cost of medical facilities [11]. On the other hand, because medical institutions’ profitability and operation are almost entirely dependent on the private sector, medical institutions must compete for service and quality of care in order to survive. In contrast to the United States’ government-driven EMR system adoption strategy [12]., South Korean medical institutions have begun to develop an information system to support the provision of high-quality medical services and to competitively promote the excellence of their medical services. As a result, the majority of institutions have customized EMR systems with unique structures and functions to meet their unique needs. As a result, one of the government’s responsibilities is to improve the quality of medical information technology systems and to alleviate the burden on medical institutions at a reasonable cost. Given that medical care is composed of systems, policies, and forms of culture, South Korea’s extremely high level of medical informatization is partially the result of cultural pressure. The state, on the other hand, is responsible for advancing the status of information technology and strengthening its capabilities. To this end, the findings of this study are sufficient to justify the utilization of superior policy resources. If this survey is repeated, it will serve as an excellent reference for evaluating the effectiveness of government policies.

### 4.3. Distinctions from Prior Survey

The most significant difference between the previous survey and 2020 is the breadth and depth of the questions. For example, we conducted an evaluation of the data exchange systems used by various government agencies and medical institutions for specific purposes. The connection between these special-purpose systems was not well known to the public during the actual investigation, as it was necessary to check each department within the hospital separately. Nonetheless, a sizable proportion—more than 95.3 percent—of tertiary institutions demonstrated that the system was connected, implying that the hospital’s data transmission cooperation with government agencies was of a high standard. The DUR system, on the other hand, was the only instance in which these data linkage results were not used solely for surveillance purposes. Government agencies must consider how to use the results of hospital data to improve the quality of medical care through appropriate feedback. We also assessed the current state of information technology systems that enable the secondary use of clinical data for research or development, using CDW, CDM, and CTMS as representative systems. The South Korean government supports the establishment of CDMs, such as the Observational Medical Outcomes Partnership (OMOP), for the purpose of facilitating and monitoring multi-institutional research. This demonstrates the critical nature of government support, particularly for CDM, as a significant number of hospitals of all sizes are implementing it with state assistance. In comparison, CDW is frequently established as an in-house system, implying that it primarily supports intra-hospital research rather than multicenter research, and external researchers rarely have access to it alone. Tertiary hospitals are increasingly establishing CDW and CDM facilities to bolster their research capabilities. In this case, it is anticipated that securing abundant research data through CDW and promoting standardization and multicenter research through CDM will have a synergistic effect. Furthermore, this study included a survey on new medical services utilizing advanced IT technologies such as IoT and 5G technology for the first time. While there were differences between institutions, it was established that hospitals were attempting to implement a variety of ICT services.

### 4.4. 4th Industrial Revolution and Smart Hospital

Based on the digital revolution of personal computers, the Internet, and information and communication technologies, the fourh Industrial Revolution is being carried out by incorporating new technologies such as artificial intelligence, big data, and the Internet of Things [13,14,15]. In terms of efficiency, economic feasibility, and the spread of the benefited class in medical care, the fourth industrial revolution, which will bring about a completely different economic and social transformation from the past, is attracting attention. Furthermore, in the face of the corona pandemic, its value as a weapon that can respond to the pandemic of infectious diseases that are essential in the life pattern of the city center close to the population is rapidly increasing. Our study, which investigates how major hospitals operating services utilizing the core technologies of the fourth industrial revolution (5G, IoT, big data, mobile, and AI) are used in actual clinical settings, would be considered extremely valuable as the first attempt at a large-scale survey. Given that the impact of rising health-care costs on national finances is growing in both developing and underdeveloped countries, as well as in developed countries, and that countermeasures must be implemented at the national level, the results of our survey could be a useful references not only in Korea, but also in other countries.

### 4.5. Limitation and Future Study

One of the primary goals of this health and medical informatization survey was to develop government policies and responses for implementing digital innovation in healthcare, as it is required in all other social sectors. It was notably challenging to find a respondent who could answer all of our questions and offer accurate responses to the questionnaire, even for a large institution with significant information infrastructure and personnel. The findings for these issues improved when the hospital had a chief information officer, a clinical informatics specialty, or a medical informatics professor. Currently, only a few OECD countries have established requirements for healthcare systems to undergo digital transformation. Denmark, Estonia, Finland, Israel, Lithuania, New Zealand, Norway, and Sweden, for example, have made significant strides but still have a long way to go. Systematically repurposing daily data, particularly for analysis and knowledge generation, continues to be a significant challenge. Additionally, digital transformation requires policy leadership and action. Institutions and organizations, not technology, are the primary impediments to developing a digital healthcare system in the twenty-first century. This entails policy actions on three fronts: (1) a digital health strategy; (2) strengthening health care data governance; and (3) institutional and operational capacity building.We learn that substantial investments have been made in five critical technologies (5G, AI, big data, IoT, and mobile). The study’s limitations include the following: first, while similar surveys have been conducted previously, it was difficult to compare them due to the surveys’ varying scopes. Second, comparisons were difficult due to the absence of comparable studies internationally. Thirdly, while the earlier surveys (2015 and 2017) were restricted to EMR construction status, there were few items that might validate time series changes. This information-gathering survey will be repeated every two or three years in the future. If it is feasible to demonstrate the serial change of the survey findings by performing repeated surveys in the future, we would like to translate the survey into English and encourage online use in terms of sharing the value of this survey and support its worldwide application. Thus, significant data may be obtained to compare the status of ICT healthcare in other countries. This study suggests that if the government and private sector collaborate to establish a more sophisticated medical ICT system and culture, a more developed future for Korea’s medical system is feasible. Furthermore, global adoption of this informatization survey has the potential to progress and improve informatization in a range of countries.

## 5. Conclusions

Our study is the most comprehensive and in-depth assessment of the current state of medical information technology ever conducted. It is expected that by conducting a detailed investigation of hospitals in South Korea that meet or exceed a certain size threshold, this data will be able to play an appropriate role in the development of the government’s digital health care strategy not only for Korea but also for all other countries. It will also be a useful resource for private institutions looking to strengthen their commitment and response plans in order to design future health care services and maintain medical information expertise.

## Figures and Tables

**Figure 1 ijerph-19-06329-f001:**
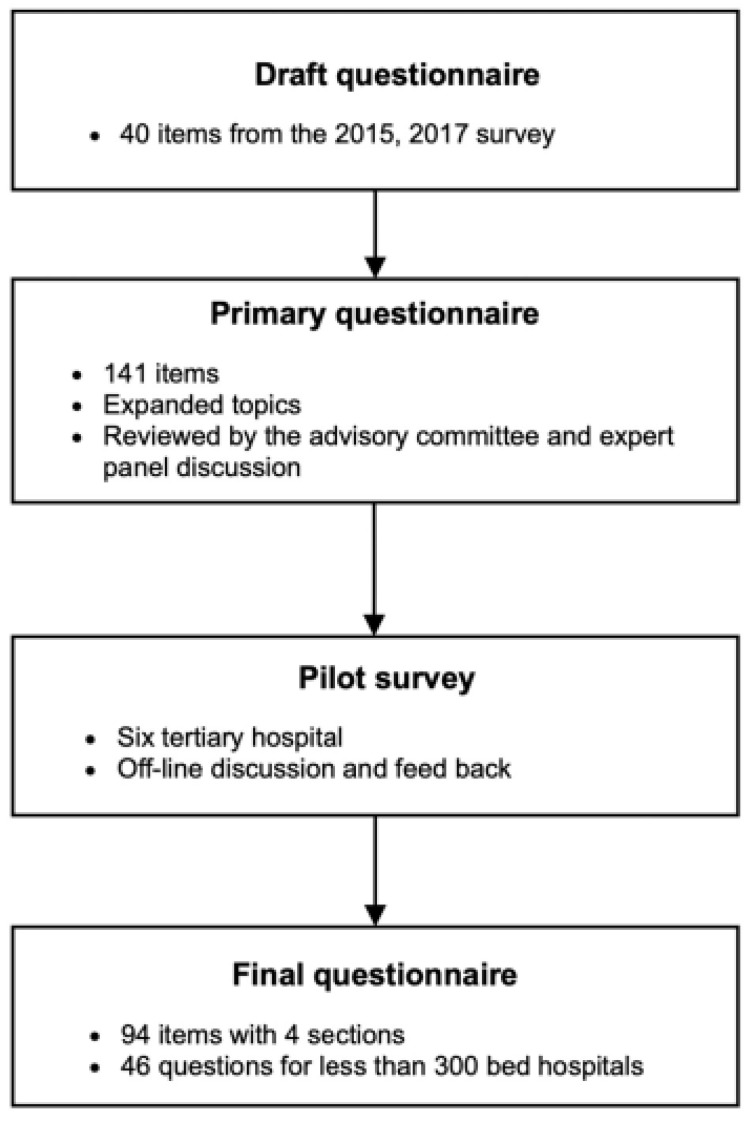
The survey questionnaire design workflow.

**Figure 2 ijerph-19-06329-f002:**
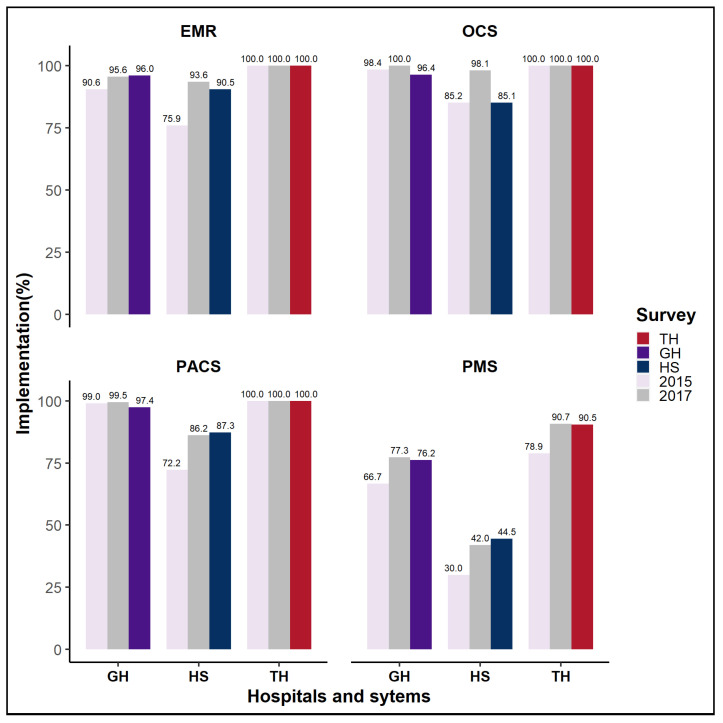
Major hospital IT systems and adoption rates in 2020. Abbreviations: EMR, electronic medical record; OCS, order communication system; PACS, picture archiving and communication system; PMS, pharmacy management system; TH, tertiary hospital; GH, general hospital; HS, hospital.

**Figure 3 ijerph-19-06329-f003:**
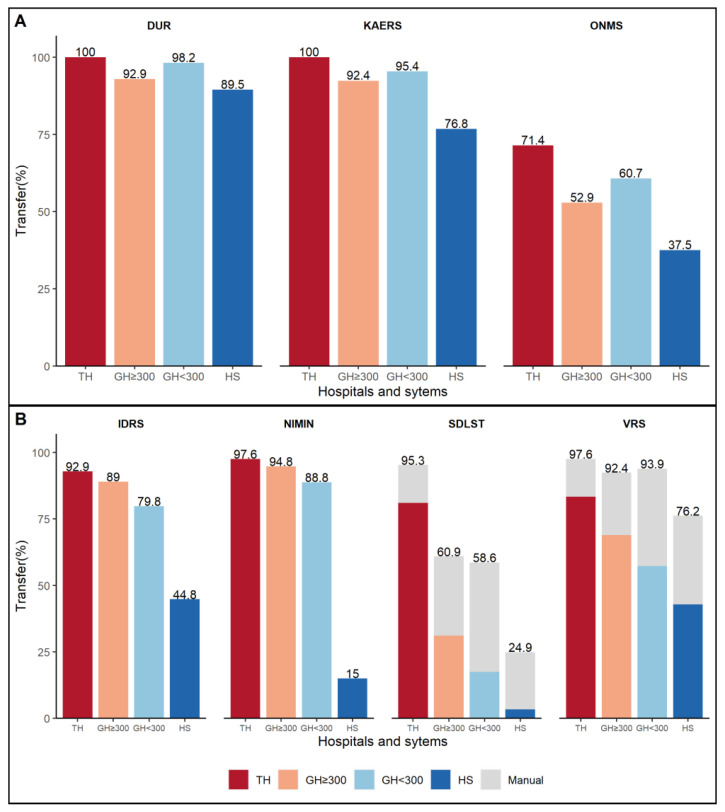
Systems for transferring clinical data to the government surveillance system (**A**,**B**). Abbreviations: TH, tertiary hospital; GH ≥ 300, general hospital with more than 300 beds; GH < 300, general hospital with less than 300 beds; HS, hospital; DUR, drug utilization review; KAERS, Korean Adverse Event Reporting System; ONMS, Opioid and Narcotics Use Management System; IDRS, Infectious Disease Report Support Program; NIMIN, National Emergency Medicine Information Network; SDLST, Life-Sustaining Medical Information Processing System; VRS, Vaccination Registration Management System.

**Figure 4 ijerph-19-06329-f004:**
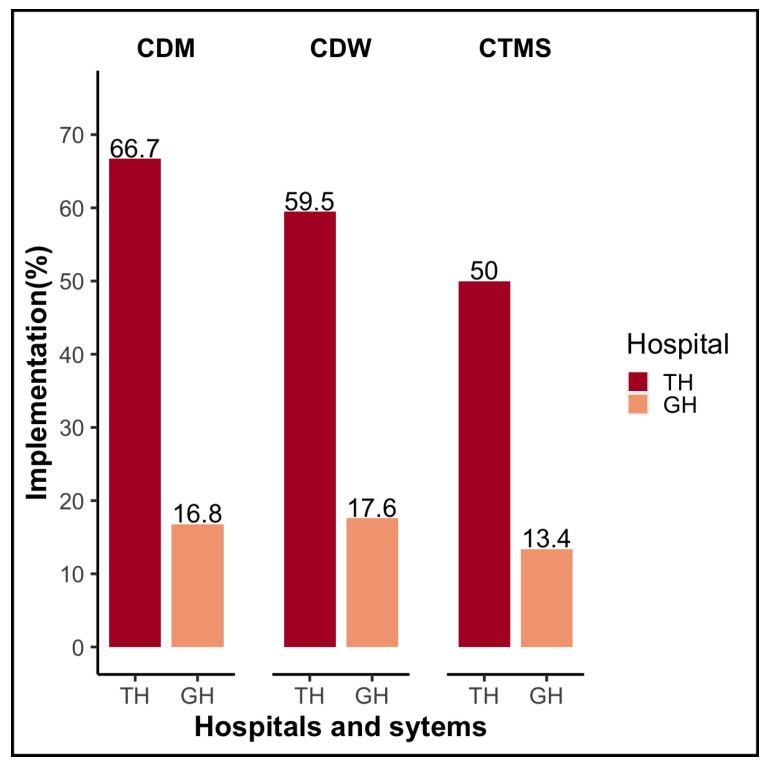
Status of CDM, CDW, and CTMS for secondary use of clinical data Abbreviations: CDM, common data model; CDW, clinical data warehouse; CTMS, clinical trial management system; TH, tertiary hospital; GH, general hospital.

**Figure 5 ijerph-19-06329-f005:**
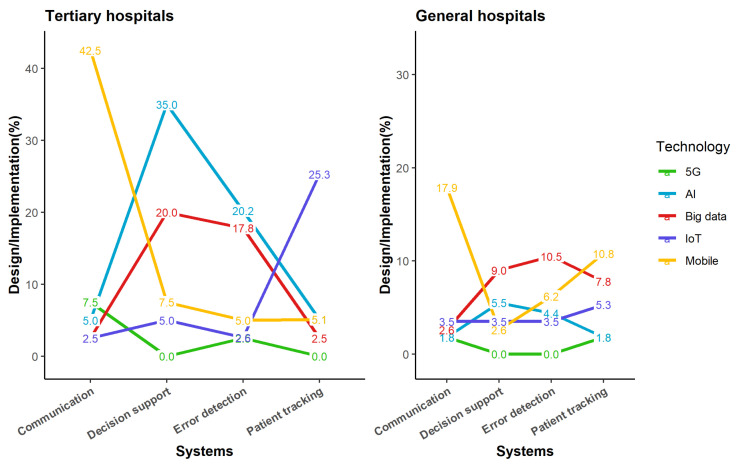
Status of core technology-based systems and/or services Abbreviations: 5G, 5th generation mobile communication; AI, artificial intelligence; IoT, internet of things.

**Table 1 ijerph-19-06329-t001:** EHealth 2020 National Profile Survey Items and Areas.

Sections	Division	Subjects
Fundamental eHealth status (Data generation)	eHealth system	IT system investment, eHealth system build and design (general status, construction history, plan, type and coverage), IT system management strategy, priority of investment, interaction of external IT system
	IT system governance	IT department status and workforce, data review committee, data quality management
	Information standard	Utilization of standard codes and terms, information exchange standard, obstacles to standard introduction
	Education and workforce	Clinical informatics education
Hospital eHealth quality (Data management)	EMR status	EMR usage status, EMR data storage management, digital signature and certificate
	EMR certification	EMR certification status, obstacles to EMR certification
	Privacy issue	Personal information protection activities, pseudonymization
	Information security	Security authentication protocol, security control, security incident response
Advanced healthcare IT system status	Advanced EMR development	EMR function improvement and advancement, CDSS design
	Information exchange	Engagement of the government-driven information exchange project
	PHR	Online patient convenience function, PHR system operation status, hospital data provision range, self-authentication method
	Smart hospital	Smart hospital service operation, smart hospital planning, smart hospital establishing policy
eHealth system for secondary use	Data utilization and sharing	Secondary use scope and regulations, patient consent system, medical data open
	IT system to support data analysis	Medical data analysis information system, CDW establishment status, CDW operation status, CDM establishment status, data sharing standard
	Infrastructure for secondary use	Clinical data utilization workforce, company participation research status, AI-based dataset construction, AI system usage status

Abbreviations: EMR, Electronic Medical Record; CDSS, Clinical Decision Support System; CDW, Clinical Data Warehouse; CDM, Common Data Model; PHR, Personal Health Record; IT, Information Technology.

**Table 2 ijerph-19-06329-t002:** Survey target hospitals and hospital classification in Korea.

Hospital Classes	Classification Criteria	Number of Hospitals	Number of Surveyed Hospitals (%)
	• Over 100 beds		
Tertiary care hospital	• Treatment of more than 20 specialized subjects	42	42 (100%)
	• Focused treatment for intractable diseases		
	• Over 100 beds		
General hospital	• 7 or more specialized subjects	311	280 (90%)
Hospital	• Over 30 beds	1431	252 (5.7%)

**Table 3 ijerph-19-06329-t003:** Status, plan, and functionality of mobile PHR between hospitals.

Adoption and Use of Mobile PHR		TH, (*n* = 42)	GH, (≥300 beds, *n* = 119)	GH, (<300 beds, *n* = 161)	Hospitals, (*n* = 252)
Use status	Active in use	61.9%	22.6%	4.4%	2.4%
	Adoption plan within 3 years	31.3%	10.8%	11.8%	10.8%
Functions	Online appointment	69.0%	24.3%	4.7%	1.9%
	Online payment	59.5%	23.6%	1.3%	0.9%
	Medical document assurance	23.8%	5.9%	2.4%	1.8%
	Providing educational material	38.1%	10.3%	1.1%	0.4%
	Medical information inquiry	64.3%	22.6%	2.5%	0.9%
	Medical information download	11.9%	0.8%	0.6%	0.4%

Abbreviations: PHR, Personal Health Record; GH, General Hospital; TH, Tertiary Hospital.

**Table 4 ijerph-19-06329-t004:** Smart hospital core technology and significant service use.

Service	Applied Status	IoT		Cloud		Big Data		Mobile		AI		5G	
		T	G	T	G	T	G	T	G	T	G	T	G
Communication between medical staff and patients	Active in use	0	0	0	0	0	0.9	22.5	4.6	0	0.9	2.5	0.9
	Plan of action	2.5	3.5	2.5	0	2.5	1.7	20.0	13.3	5.0	0.9	5.0	0.9
Medical service design and resource utilization	Active in use	22.5	1.8	0	0	0	0.9	5.0	0	2.5	0.9	0	0
	Plan of action	15.0	2.8	0	0	2.5	3.6	10.0	3.6	7.5	0.9	7.5	1.8
Medication error detection	Active in use	2.6	0.9	0	0	12.8	1.7	0	0	7.7	0.9	0	0
	Plan of action	0	2.6	2.5	0	5.0	8.8	5.0	6.2	12.5	3.5	2.5	0
AI-based decision support model	Active in use	2.5	0.9	0	0	5.0	0.9	0	0	17.5	0.9	0	0
	Plan of action	2.5	2.6	0	2.7	15.0	8.1	7.5	2.6	17.5	5.5	0	0
AI-based digital therapeutics	Active in use	0	0.9	0	0	4.9	0.9	7.3	0.0	17.1	2.8	0	0.9
	Plan of action	2.5	1.9	2.5	0.9	10.0	6.4	5.0	4.6	22.5	6.4	5.0	0.9
Early confrontation to infectious crisis	Active in use	0.0	2.8	0	0	5.1	0.9	12.8	2.7	0	0	0	0.9
	Plan of action	10.0	1.8	0	0	5.0	4.4	5.0	6.3	5.0	0	2.5	0.9
At-risk and emergency patients tracking	Active in use	12.8	1.8	0	0	0	1.7	2.6	2.7	2.6	0.9	0	0.9
	Plan of action	12.5	3.5	0	0	2.5	6.1	2.5	8.1	2.5	0.9	0	0.9
Hazard alert and monitoring	Active in use	2.6	3.7	0	0	5.1	1.7	10.3	3.6	2.6	0	7.7	0.9
	Plan of action	10.0	4.4	0	0.9	0	0.9	7.5	11.7	10.0	2.7	7.5	0

Abbreviations: G, General hospital; T, Tertiary hospital; IoT, internet of things; AI, artificial intelligence.

## Data Availability

Not applicable.

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
