# Peer review of "Digital Health Profile of South Korea: A Cross Sectional Study"

_ijerph, 2022, doi:10.3390/ijerph19106329_

Round 1

Reviewer 1 Report

The authors  designed and performed a nationwide survey of medical institutions’ ICT systems To assess the current state of healthcare’s information and communication technology (ICT) systems and to envision the future of systems and advanced ICT healthcare services,

They performed the study  from November 16th through December 11th, 2020, applied to  42 tertiary hospitals, 311 general hospitals, and 1,431 hospital locations countrywide. (3)

Their outcome highlight that:

 (A)  since 2015, most hospitals have implemented electronic medical record (EMR) systems (90.5 percent of hospitals, which is the smallest unit, and 100 percent of tertiary hospitals).

(B) The rate of implementation of personal health records (PHRs) varied significantly between 61.9 percent and 2.4 percent, depending on the size of the hospital.

(C)Hospitals have implemented around three to seven government-sponsored information/data transmission and receiving systems for statistical or investigative objectives.

(D) For secondary usage of medical data, more than half of tertiary hospitals have implemented a clinical data warehouse or shared data model.

(E) New service establishments utilizing modern medical technologies such as artificial intelligence or lifelogging were scarce and in the planning stages.

I agree with the conclusions of the authors reporting that the study demonstrates a high level of informatization in Korean medical institutions, implying that high-quality next-generation medical services can be developed and implemented in the future with sufficient government assistance.

This is a nice, interesting and well written piece.

With a pure academic spirit I offer the authors the following suggestions:

  1. Better explicit the purpose.
  2. Put a table with acronyms.
  3. The survey is electronic or paper based?
  4. If it is electronic please give some information. If it is not electronic discuss if it is planned an electronic development for the future
  5. Methods and results are correctly arranged into paragraphs. Please introduece these paragraphs with a few sentences
  6. Check the resolution and quality of the tables and figures
  7. Put a conclusion section highlighting the contribution of the study.

Author Response

Response to Reviewer 1 Comments

We sincerely appreciate the reviewer's intriguing remarks. Despite your hectic schedule, you have read our paper thoroughly and provided detailed suggestions for improvement, allowing us to produce a higher-quality manuscript. Following is a point-by-point breakdown of the revisions and responses to the reviewers' comments.

Point 1: Better explicit the purpose.

Response 1: I completely agree with your point of view. Because the goals we were attempting to achieve in our study were so broad and comprehence, it appears that the systematic description of these goals in the paper was inadequate. In response to your comments, we revised the first sentence of the Abstract, and we structured and specifically described the research goals in the Introduction section as follows.

In the Abstract (line 21-23), “Background: For future national digital healthcare policy development, it is vital to collect baseline data on the infrastructure and services of medical institutions' information and communication technology (ICT). To assess the state of medical ICT across the nation, we devised and administered a comprehensive digital healthcare survey to medical institutions across the nation.”

Point 2: Put a table with acronyms.

Response 2: We have added abbreviations in Table 3. The following are the revised table 3 and texts.

Point 3 & 4 : The survey is electronic or paper based?

If it is electronic please give some information. If it is not electronic discuss if it is planned an electronic development for the future.

Response 3 & 4: We conducted a paper-based survey and published the questions and findings in Korean. As the reviewer stated, our survey questions will be useful to people in different nations. We planned to do this survey every other year or every three years in the future. In such instance, an English translation for worldwide use will be created. These are now in the Discussion section.

In the Discussion section, line 349-357

: This information-gathering survey will be repeated every two or three years in the future. If it is feasible to demonstrate the serial change of the survey findings by performing repeated surveys in the future, we would like to translate the survey into English and encourage online use in terms of sharing the value of this survey and support its worldwide application. Thus, significant data may be obtained to compare the status of ICT healthcare in other countries. This study suggests that if the government and private sector collaborate to establish a more sophisticated medical ICT system and culture, a more developed future for Korea's medical system is feasible. Furthermore, global adoption of this informatization survey has the potential to progress and improve informatization in a range of countries.

Point 5: Methods and results are correctly arranged into paragraphs. Please introduece these paragraphs with a few sentences

Response 5: I agree with the reviewers' remarks. We've included a quick description to the beginning of Methods and Results to help our readers better understand the findings of this study. Additions include the following lines of text.

In the Method section, line 63-74

This research aimed to examine the medical ICT infrastructure and service status of as many hospitals as feasible. To this purpose, we intended to go beyond the limits of the national EMR system building status surveys conducted in 2015 and 2017. IT systems to facilitate secondary use of medical data, additional systems beyond EMR, such as PHR and linkage with government surveillance systems, the status of medical services leveraging emerging technologies such as artificial intelligence, etc., were all newly included items in the 2020 survey. Expert consensus established the scope of the survey so that the results might serve as fundamental data for future health information governance and policy development. The study was conducted in stages, beginning with establishing the scope of the hospital to be surveyed, constructing a survey by field, running a prototype survey and getting comments, confirming the final survey questionnaire, and then completing the actual survey.

In the Result section, line 119-123

: After reviewing the total response rate, we subdivided and examined the response findings based on the primary scope of the survey. There were numerous complex inquiries in four areas, and the investigation's extent varied based on the size of the hospital; therefore, we separated the results into six subjects and discussed each one separately to facilitate comprehension.

Point 6: Check the resolution and quality of the tables and figures

Response 6: Thank you for the reviewer's comments. As I submitted my thesis in Latex, the resolution was automatically reduced. High resolution figures are attached separately to avoid problems in future publications.

Point 7: Put a conclusion section highlighting the contribution of the study.

Response 7: We appreciate the appropriate comments from the reviewers. We have added a Conclusion section to communicate to our readers the contribution of this paper to this field. The added content is as follows.

Conclusion, line 359-365

: Our study is the most comprehensive and in-depth assessment of the current state of medical information technology ever conducted. It is expected that by conducting a detailed investigation of hospitals in South Korea that meet or exceed a certain size threshold, this data will be able to play an appropriate role in the development of the government's digital health care strategy not only for Korea but also for all other countries. It will also be a useful resource for private institutions looking to strengthen their commitment and response plans in order to design future health care services and maintain medical information expertise.

Reviewer 2 Report

The work exposes an important challenge in health care, on the issue of the level of informatization in hospitals in Korea. In particular, the authors designed and conducted a nationwide survey of the ICT systems of medical institutions. In their work, they affirm that the informatization of hospitals is extremely important since, for effective infection control, as in the case of the global coronavirus disease, a clear and accurate information exchange is required for rapid decision-making. The work highlights the broad participation of experts, government and public organizations in the health area in the design of the survey, with which expert opinions are incorporated and the quality of the survey is improved. In addition, of the wide application of the survey in the hospitals of Korea. The work is of acceptable quality, but it can be substantially improved if the following recommendations are followed:

In the Abstract section, they must make clear the contributions of their work. In addition, the first sentence of the summary is not well written and does not make the message clear.

Although the authors explain that they used several instruments to apply the survey, they do not describe how they developed the process and what the most critical problems were.

Table 4 should be explained a little more, for example, what the value of each column means.

Figure 5 should be explained a bit more, as it is important to know the authors' conclusions.

On line 26, in the introduction section, you don't have the referential citation “[?]”

In line 192 and 197 the number of the reference figure is missing.

It is desirable that they increase the Reference Section, with a couple more references.

Author Response

Response to Reviewer 2 Comments

We sincerely appreciate the reviewer's intriguing remarks. Despite your hectic schedule, you have read our paper thoroughly and provided detailed suggestions for improvement, allowing us to produce a higher-quality manuscript. Following is a point-by-point breakdown of the revisions and responses to the reviewers' comments.

Point 1: In the Abstract section, they must make clear the contributions of their work. In addition, the first sentence of the summary is not well written and does not make the message clear.

Response 1: I strongly agree with the reviewers' comments. We revised the first sentence of the Abstract section to better communicate the purpose and contribution of our study to readers. We also added a summary of this study's contribution to the last sentence of the Abstract section. The following are the edited contents.

In the Abstract (line 1-4), “Background: For future national digital healthcare policy development, it is vital to collect baseline data on the infrastructure and services of medical institutions' information and communication technology (ICT). To assess the state of medical ICT across the nation, we devised and administered a comprehensive digital healthcare survey to medical institutions across the nation.”

In the Abstract (line 15-19), “This study shows that the level of digitalization in Korean medical institutions is significant, despite the fact that the development and spending in ICT infrastructure and services provided by individual institutions imposes a significant cost. This illustrates that, in the face of a pandemic, strong government backing and policymaking are essential to activate ICT-based medical services and efficiently use medical data.

Point 2: Although the authors explain that they used several instruments to apply the survey, they do not describe how they developed the process and what the most critical problems were.

Response 2: We appreciate the reviewer's feedback. A overview of the research methodologies has been added to the Methods section to assist readers in understanding how the survey research process was constructed. And the most challenging issue encountered during the research process was added to the Discussion section. The following are the revised texts.

In the Methods section, line 63~74

: This research aimed to examine the medical ICT infrastructure and service status of as many hospitals as feasible. To this purpose, we intended to go beyond the limits of the national EMR system building status surveys conducted in 2015 and 2017. IT systems to facilitate secondary use of medical data, additional systems beyond EMR, such as PHR and linkage with government surveillance systems, the status of medical services leveraging emerging technologies such as artificial intelligence, etc., were all newly included items in the 2020 survey. Expert consensus established the scope of the survey so that the results might serve as fundamental data for future health information governance and policy development. The study was conducted in stages, beginning with establishing the scope of the hospital to be surveyed, constructing a survey by field, running a prototype survey and getting comments, confirming the final survey questionnaire, and then completing the actual survey.

In the Discussion section, line 329-333

: It was notably challenging to find a respondent who could answer all of our questions and offer accurate responses to the questionnaire, even for a large institution with significant information infrastructure and personnel. The findings for these issues improved when the hospital had a chief information officer, a clinical informatics specialty, or a medical informatics professor.

Point 3&4 Table 4 should be explained a little more, for example, what the value of each column means. Figure 5 should be explained a bit more, as it is important to know the authors' conclusions.

Response 3&4: We appreciate the reviewers' insightful comments. The following are more detailed explanations for Table 4 and Figure 5.

In the Results section, line 213-242

We evaluated how and how much each of the five fundamental technologies–internet of things (IoT), big data, mobile, artificial intelligence (AI), and fifth-generation technological standard (5G)–was used in each hospital and their future investment plans. These five technologies were included in the survey because the Korean Ministry of Health and Welfare selected them as new technologies requiring active investment for the future transition to digital medicine. To confirm the usage status of these technologies in-depth, we investigated how these technologies are being used or are planned to be used for eight major services derived from expert consensus. The survey asked respondents about their experiences with these technologies in eight distinct service areas: 1) communication between medical staffs and patients, 2) Medical service design and resource utilization, 3) Medication error detection, 4) AI-based decision support mode, 5) AI-based digital therapeutics, 6) Early confrontation to infectious crisis, 7) At-risk and emergency patients tracking, and 8) Hazard alert and monitoring. The utilization status of each technology for each service was evaluated and categorized as "active use" or "action plan," and the findings were analyzed by dividing by the hospital's size. Table 4 summarizes these domains and the extent to which the five key technologies are applied. Four of the eight services confirmed an instance of a differentiating characteristic for five technologies:  Communication between medical staffs and patients (Communication), AI-based decision support model (Decision support), Medication error detection (Error detection), At-risk and emergency patients tracking (Patient tracking). As indicated in Figure 5, there are significant disparities between the sorts of services and the fundamental technologies believed to be required to supply these services. Specifically, expected benefits such as the active use of mobile technologies for communication and IoT technology for patient tracking were demonstrated. In addition, it was established that big data and artificial intelligence technology displayed a close correlation.

According to respondents, mobile technology is the most critical technology for communication services, while AI is the most core technology for decision support services; AI and big data are the most pivotal technologies for error detection, while IoT is the most powerful technology for patient tracking, as illustrated in Figure 5. This attitude, however, varied according to hospital size; opinion about the importance of mobile technology was generally more favorable in general hospitals than in tertiary hospitals.

Point 5&6: On line 26, in the introduction section, you don't have the referential citation “[?]”. In line 192 and 197 the number of the reference figure is missing.

Response 5&6: Thank you very much. A typo occurred while writing Latex. Each error has been rectified.

Point 7: It is desirable that they increase the Reference Section, with a couple more references.

Response 7: I agree with the comments given by the reviewers. We supplemented our preliminary research in the Discussion section, adding some references and supplementing related discussions. The edited contents are as follows.

In the Discussion section, line 309~325

4th Industrial Revolution and Smart Hospital

Based on the digital revolution of personal computers, the Internet, and information and communication technologies, the 4th Industrial Revolution is being carried out by incorporating new technologies such as artificial intelligence, big data, and the Internet of Things \cite{ref-journal13, ref-journal14, ref-journal15}. In terms of efficiency, economic feasibility, and the spread of the benefited class in medical care, the fourth industrial revolution, which will bring about a completely different economic and social transformation from the past, is attracting attention. Furthermore, in the face of the corona pandemic, its value as a weapon that can respond to the pandemic of infectious diseases that are essential in the life pattern of the city center close to the population is rapidly increasing. Our study, which investigates how major hospitals operating services utilizing the core technologies of the fourth industrial revolution (5G, IoT, big data, mobile, and AI) are used in actual clinical settings, would be considered extremely valuable as the first attempt at a large-scale survey. Given that the impact of rising health-care costs on national finances is growing in both developing and underdeveloped countries, as well as in developed countries, and that countermeasures must be implemented at the national level, the results of our survey could be a useful references not only in Korea, but also in other countries.

Reviewer 3 Report

  1. What is the main question addressed by the research?

This study focuses on the implementation of various data repository systems, including CDW, CDM and CTMS, for the secondary use of clinical data. The authors attempted to assess the current state of healthcare information and communication technology systems and to envisage the future of ICT systems and advanced healthcare services.

  1. Do you consider the topic original or relevant in the field? Does it address a specific gap in the field?

The assessment of the state of healthcare information and communication technology systems is an important element of public safety, since modern IT developments have a positive impact on the development of new ways of organizing healthcare for the population, and therefore contributes to the preservation of the nation's health. In this context the paper is relevant.

  1. What does it add to the subject area compared with other published material?

This paper attempted to assess the functioning of eHealth 2020 in South Korea by examining four key categories: fundamental eHealth status, hospital eHealth quality, advanced healthcare IT system status, and eHealth system for secondary use. It was also meant to measure the rate of implementation of various data repository systems for secondary use of clinical data.

  1. What specific improvements should the authors consider regarding the methodology? What further controls should be considered?

1) The article lacks clarity in defining the purpose of the study. The authors correct the purpose of the study three times:
Purpose 1 (line 1-2): To assess the current state of healthcare’s information and communication technology (ICT) systems and to envision the future of systems and advanced ICT healthcare services.
Purpose 2 (line 210-213): The purpose of this study was to measure the rate of implementation of various data repository systems, including CDW, CDM, and CTMS, for secondary use of clinical data, and this was the first attempt to examine clinical research system use on a national scale.
Purpose 3 (line 267-268) One of the primary goals of this health and medical informatization survey was to develop government policies and responses for implementing digital innovation in healthcare, as it is required in all other social sectors.
If there are several purposes, why aren't they structured?

2) The review of publications on the topic of the study is very limited. This is also confirmed by the number of literature sources used.

3) In the section on materials and methods, the period of the study should have been specified in accordance with the purpose of the study.

4) The paper is a promising material, but it lacks the application of advanced statistical tools for the analysis of respondents' statements by questionnaires. The use of advanced statistical methods would probably allow to draw more interesting conclusions.

  1. Are the conclusions consistent with the evidence and arguments
    presented and do they address the main question posed?

Due to the vagueness of the research purpose, it cannot be stated that it has been achieved in the final part of the paper.

  1. Are the references appropriate?

The literature cited in this paper corresponds to the current state of research on the issues discussed, but it is worth extending the review of publications considerably by looking at the views of economists in the field under study.

  1. Please include any assitional comments on the tables and figures.

Figure 2 is not formed correctly enough. From the proposed legend it is difficult to understand where the data are from 2020, and that the data for 2015 and 2017 are divided by levels of healthcare service.

From the text of the paper and the labels in Fig. 3 it is not clear which period the data refer to.

  1. Additional comments or continuations of above sections.

If the purpose of this study was "... to measure the rate of implementation of various data repository systems" (lines 210-211)", then why the data for the period 2015, 2017, 2020 are shown exclusively in Figure 2? Furthermore, the uneven intervals of the study may distort the conclusions drawn.

Author Response

Response to Reviewer 3 Comments

We sincerely appreciate the reviewer's intriguing remarks. Despite your hectic schedule, you have read our paper thoroughly and provided detailed suggestions for improvement, allowing us to produce a higher-quality manuscript. Following is a point-by-point breakdown of the revisions and responses to the reviewers' comments.

Point 1: What is the main question addressed by the research?

This study focuses on the implementation of various data repository systems, including CDW, CDM and CTMS, for the secondary use of clinical data. The authors attempted to assess the current state of healthcare information and communication technology systems and to envisage the future of ICT systems and advanced healthcare services.

Response 1: We are grateful for the reviewer's rating of our study.

Point 2: Do you consider the topic original or relevant in the field? Does it address a specific gap in the field?

The assessment of the state of healthcare information and communication technology systems is an important element of public safety, since modern IT developments have a positive impact on the development of new ways of organizing healthcare for the population, and therefore contributes to the preservation of the nation's health. In this context the paper is relevant.

Response 2: We are grateful to the reviewers for their thoughtful appraisal of the study's subject matter.

Point 3 : What does it add to the subject area compared with other published material?

This paper attempted to assess the functioning of eHealth 2020 in South Korea by examining four key categories: fundamental eHealth status, hospital eHealth quality, advanced healthcare IT system status, and eHealth system for secondary use. It was also meant to measure the rate of implementation of various data repository systems for secondary use of clinical data.

Response 3: We thank the reviewers for confirming the scope of the subject matter of this study.

Point 4-1: What specific improvements should the authors consider regarding the methodology? What further controls should be considered?

1) The article lacks clarity in defining the purpose of the study. The authors correct the purpose of the study three times:
Purpose 1 (line 1-2): To assess the current state of healthcare’s information and communication technology (ICT) systems and to envision the future of systems and advanced ICT healthcare services.
Purpose 2 (line 210-213): The purpose of this study was to measure the rate of implementation of various data repository systems, including CDW, CDM, and CTMS, for secondary use of clinical data, and this was the first attempt to examine clinical research system use on a national scale.
Purpose 3 (line 267-268) One of the primary goals of this health and medical informatization survey was to develop government policies and responses for implementing digital innovation in healthcare, as it is required in all other social sectors.
If there are several purposes, why aren't they structured?

Response 4-1: I agree and respect the reviewer's points of view. Our investigation encompassed ICT infrastructure used in clinical practice, research, and secondary use, as well as future digital health care needs and services. As a result, the reviewers remarked that the research objectives and important questions were rather disorganized. Several modifications were made to the text in order to make the major questions and obstacles of this study more clear to the reader.

By altering the first and last phrases of the Abstract, the objective and contribution of our thesis are described as follows.

In the Abstract (line 1-4), “Background: For future national digital healthcare policy development, it is vital to collect baseline data on the infrastructure and services of medical institutions' information and communication technology (ICT). To assess the state of medical ICT across the nation, we devised and administered a comprehensive digital healthcare survey to medical institutions across the nation.”

In the Abstract (line 15-19), “This study shows that the level of digitalization in Korean medical institutions is significant, despite the fact that the development and spending in ICT infrastructure and services provided by individual institutions imposes a significant cost. This illustrates that, in the face of a pandemic, strong government backing and policymaking are essential to activate ICT-based medical services and efficiently use medical data.

altering the first and last phrases of the Abstract, the objective and contribution of our thesis are described as follows.

In the Method section, (line 63-74),

: This research aimed to examine the medical ICT infrastructure and service status of as many hospitals as feasible. To this purpose, we intended to go beyond the limits of the national EMR system building status surveys conducted in 2015 and 2017. IT systems to facilitate secondary use of medical data, additional systems beyond EMR, such as PHR and linkage with government surveillance systems, the status of medical services leveraging emerging technologies such as artificial intelligence, etc., were all newly included items in the 2020 survey. Expert consensus established the scope of the survey so that the results might serve as fundamental data for future health information governance and policy development. The study was conducted in stages, beginning with establishing the scope of the hospital to be surveyed, constructing a survey by field, running a prototype survey and getting comments, confirming the final survey questionnaire, and then completing the actual survey.

Point 4-2: The review of publications on the topic of the study is very limited. This is also confirmed by the number of literature sources used.

Response 4-2: I agree with the reviewers' observations. In the Discussion section, we expanded on our preliminary research by adding references and expanding on related discussions. The following are the edited contents.

In the Discussion section, line 309-325

4th Industrial Revolution and Smart Hospital

Based on the digital revolution of personal computers, the Internet, and information and communication technologies, the 4th Industrial Revolution is being carried out by incorporating new technologies such as artificial intelligence, big data, and the Internet of Things (13-15). In terms of efficiency, economic feasibility, and the spread of the benefited class in medical care, the fourth industrial revolution, which will bring about a completely different economic and social transformation from the past, is attracting attention. Furthermore, in the face of the corona pandemic, its value as a weapon that can respond to the pandemic of infectious diseases that are essential in the life pattern of the city center close to the population is rapidly increasing. Our study, which investigates how major hospitals operating services utilizing the core technologies of the fourth industrial revolution (5G, IoT, big data, mobile, and AI) are used in actual clinical settings, would be considered extremely valuable as the first attempt at a large-scale survey. Given that the impact of rising health-care costs on national finances is growing in both developing and underdeveloped countries, as well as in developed countries, and that countermeasures must be implemented at the national level, the results of our survey could be a useful references not only in Korea, but also in other countries.

Point 4-3: In the section on materials and methods, the period of the study should have been specified in accordance with the purpose of the study.

Response 4-3: Thank you for your insightful feedback. Only the survey time is provided in our manuscript (16 November to 11 December 2020), thus we represented in the Methods section alongside the study design and pilot test period. The following are the contents after editing.

In the Method section, line 106-117

Due of the questionnaire's extensive reach and depth, responding to the survey was challenging. To address this obstacle, we used two distinct survey methodologies deliberately. The KHIA and KHMIMA collaborated on the field investigation for the large-scale hospitals (more than 300 beds). Interviewers were trained by medical information professionals from these organizations. In addition, two online briefing sessions were held for respondents participating in the research response at each hospital. With fewer than 300 beds, Gallop Korea used a combination of survey methods, including online surveys, fax/e-mail surveys, and in-person interviews (using tablet computers or paper questionnaires). The study design and pilot survey were carried out from September to October 2020, and the online briefing session was held on November 16, 2020. The final survey was conducted from November 16, 2020, to December 11, 2020, with some supplementary surveys continuing until December 31, 2020.

Point 4-4: The paper is a promising material, but it lacks the application of advanced statistical tools for the analysis of respondents' statements by questionnaires. The use of advanced statistical methods would probably allow to draw more interesting conclusions.

Response 4-4:

Thank you for the feedback from the reviewer. Our authors also spent a lot of time considering what important conclusions could be made from our findings. And data on the differences between hospitals that are taking a risky posture as an advanced ICT hospital (for example, hospitals that actively push the use of technologies like AI, mobile, big data, and IoT) and hospitals that aren't.

 I wanted to see how it compared to a model. First, the total outcome was examined for action, action + plan, and plan, with logistic regression analysis applied to action and action + plan for six specific elements. There were few significant criteria, therefore only a couple were chosen. Because there is a lot of space for interpretation with this result, we did not offer it as the major result in the study.

Point 5: Are the conclusions consistent with the evidence and arguments
presented and do they address the main question posed?

Due to the vagueness of the research purpose, it cannot be stated that it has been achieved in the final part of the paper.

Response 5: Thank you for your positive feedback.

Point 6: Are the references appropriate?

The literature cited in this paper corresponds to the current state of research on the issues discussed, but it is worth extending the review of publications considerably by looking at the views of economists in the field under study.

Response 6:. I agree with the reviewers' observations. In the Discussion section, we expanded on our preliminary research by adding references and expanding on related discussions. Based on the reviewers' opinions, studies conducted from the standpoint of economists were added as references. The following are the edited contents.

In the Discussion section, line 309-325

4th Industrial Revolution and Smart Hospital

Based on the digital revolution of personal computers, the Internet, and information and communication technologies, the 4th Industrial Revolution is being carried out by incorporating new technologies such as artificial intelligence, big data, and the Internet of Things (13-15). In terms of efficiency, economic feasibility, and the spread of the benefited class in medical care, the fourth industrial revolution, which will bring about a completely different economic and social transformation from the past, is attracting attention. Furthermore, in the face of the corona pandemic, its value as a weapon that can respond to the pandemic of infectious diseases that are essential in the life pattern of the city center close to the population is rapidly increasing. Our study, which investigates how major hospitals operating services utilizing the core technologies of the fourth industrial revolution (5G, IoT, big data, mobile, and AI) are used in actual clinical settings, would be considered extremely valuable as the first attempt at a large-scale survey. Given that the impact of rising health-care costs on national finances is growing in both developing and underdeveloped countries, as well as in developed countries, and that countermeasures must be implemented at the national level, the results of our survey could be a useful references not only in Korea, but also in other countries.

Point 7: Please include any assitional comments on the tables and figures.

Figure 2 is not formed correctly enough. From the proposed legend it is difficult to understand where the data are from 2020, and that the data for 2015 and 2017 are divided by levels of healthcare service.

From the text of the paper and the labels in Fig. 3 it is not clear which period the data refer to.

Response 7: I concur with the reviewer's evaluation. The colors in Figure 2 have been modified to make it simpler to differentiate between 2015 and 2017. In the case of Figure 3, a related explanation has been included to the result section text.

In the Result section, line

This topic was first evaluated in the 2020 survey.

Point 8: Additional comments or continuations of above sections.

If the purpose of this study was "... to measure the rate of implementation of various data repository systems" (lines 210-211)", then why the data for the period 2015, 2017, 2020 are shown exclusively in Figure 2? Furthermore, the uneven intervals of the study may distort the conclusions drawn.

Response 8:. Thank you a great deal. The surveys conducted in 2015 and 2017 were limited examinations of EMR implementations. The scope of the 2020 study has been considerably enlarged, with little overlap with earlier investigations. Consequently, only the elements illustrated in Figure 2 were accessible for time series comparison. The Discussion section also addressed the limits of the study's conclusion as a result of the 2020 survey's different scope.

In the Discussion section, line

Thirdly, while the earlier surveys (2015 and 2017) were restricted to EMR construction status, there were few items that might validate time series changes.

Round 2

Reviewer 3 Report

The structure of the paper has improved and the comments have been taken into account for the most part. However, the number of literature sources used is still debatable.

I leave the decision regarding publication to the editor.